# A Novel Non-Contact Detection and Identification Method for the Post-Disaster Compression State of Injured Individuals Using UWB Bio-Radar

**DOI:** 10.3390/bioengineering10080905

**Published:** 2023-07-30

**Authors:** Ding Shi, Fulai Liang, Jiahao Qiao, Yaru Wang, Yidan Zhu, Hao Lv, Xiao Yu, Teng Jiao, Fuyuan Liao, Keding Yan, Jianqi Wang, Yang Zhang

**Affiliations:** 1Department of Medical Electronics, School of Biomedical Engineering, Air Force Medical University, Xi’an 710032, China; sd_411276364@163.com (D.S.); liangfulai@fmmu.edu.cn (F.L.); qiaojiahao2021@163.com (J.Q.); wyaru0328@163.com (Y.W.); zhuyidan1210@163.com (Y.Z.); fmmulvhao@fmmu.edu.cn (H.L.); yuxiao@fmmu.edu.cn (X.Y.); jiaoteng@fmmu.edu.cn (T.J.); 2Shaanxi Provincial Key Laboratory of Bioelectromagnetic Detection and Intelligent Perception, Air Force Medical University, Xi’an 710032, China; 3Department of Biomedical Engineering, School of Electronic and Information Engineering, Xi’an Technological University, Xi’an 710032, China; liaofuyuan1024@163.com (F.L.); yankeding168@163.com (K.Y.)

**Keywords:** bio-radar, crushing injury, ultra-wideband, variational modal decomposition, convolutional neural network

## Abstract

Building collapse leads to mechanical injury, which is the main cause of injury and death, with crush syndrome as its most common complication. During the post-disaster search and rescue phase, if rescue personnel hastily remove heavy objects covering the bodies of injured individuals and fail to provide targeted medical care, ischemia-reperfusion injury may be triggered, leading to rhabdomyolysis. This may result in disseminated intravascular coagulation or acute respiratory distress syndrome, further leading to multiple organ failure, which ultimately leads to shock and death. Using bio-radar to detect vital signs and identify compression states can effectively reduce casualties during the search for missing persons behind obstacles. A time-domain ultra-wideband (UWB) bio-radar was applied for the non-contact detection of human vital sign signals behind obstacles. An echo denoising algorithm based on PSO-VMD and permutation entropy was proposed to suppress environmental noise, along with a wounded compression state recognition network based on radar-life signals. Based on training and testing using over 3000 data sets from 10 subjects in different compression states, the proposed multiscale convolutional network achieved a 92.63% identification accuracy. This outperformed SVM and 1D-CNN models by 5.30% and 6.12%, respectively, improving the casualty rescue success and post-disaster precision.

## 1. Introduction

Natural disasters and local wars have become frequent in recent years. According to statistical data, the proportion of mechanical injuries to the human body caused by heavy object crushing after disasters and conflicts has reached 48.37% [1]. Crush syndrome caused by external heavy objects (such as collapsed buildings) is the most common complication of mechanical injury caused by heavy object crushing and usually refers to ischemic necrosis of muscle tissue [2,3]. If the compression of an injured person is casually released without careful diagnosis during a rescue, local lesions and ischemia-reperfusion injury can result in rhabdomyolysis of the injured person [4]. In severe cases, diffuse intravascular coagulation or acute respiratory distress syndrome may also occur, leading to multiple organ failure, and ultimately shock and death of the injured person [5,6]. Compared with traditional non-contact post-disaster search and rescue devices, such as visible light or thermal imaging [7], bio-radar can detect vital signs, such as breathing and a heartbeat, caused by human cardiopulmonary activity at longer distances and behind obstacles [8,9]. It has become one of the most promising technologies for disaster rescue detection [10,11,12,13,14,15,16,17].

Research on the application of bio-radar in the field of vital sign detection in casualties can be traced back to the 1980s. KunMu Chen from Michigan State University in the United States used an X-band bio-radar to detect human subjects behind cinder brick walls [18]. This study explored a method that can confirm location information and monitor the physiological information of wounded soldiers in a non-contact manner to maximize rescue efficiency. With the continuous advancement of research, this non-contact life-detection method has received increasing attention from researchers worldwide. Liu from the University of Connecticut conducted an electromagnetic simulation of human heart and lung signals based on the finite-difference time-domain (FDTD) method and carried out experiments using a bio-radar with a central frequency of 1 GHz. Through data processing combined with the Hilbert Huang (HHT) algorithm, they successfully recognized and distinguished the three respiratory characteristics of different subjects after obstacles [19]. Zhang from the Air Force Medical University of China used a dual-frequency bio-radar for wall-penetrating vital sign detection and adaptive filtering algorithms to filter out respiratory interference outside the detection background, further improving the detection accuracy of the bio-radar [20]. Wu used an ultra-wideband linear array to detect the respiratory signal of a human body under rubble and estimated the human respiratory movement and position of the human body behind obstacles [21]. Yan from the Nanjing University of Science and Technology proposed a time-varying vital sign separation algorithm based on a variational modal decomposition, which is used to suppress interference and aliasing in the detection of multi-target radar vital signals behind walls. This method was verified to be effective at extracting respiration signals from multiple human targets [22]. Researchers have also explored the vital status information of these targets in addition to being interested in the number and location of human targets behind obstacles. In 2019, Yin et al. from the Air Force Medical University conducted a study on the changes in the vital statuses of animals in a scenario involving obstacles and blood loss using bio-radar. This study made progress in distinguishing the vital statuses of animals in terms of their instantaneous frequency characteristics and energy ratios [23]. Afterward, Ma proposed an early warning method for biological depletion periods based on “M”-shaped radar breathing signals. In this study, the physiological signals of animals under water and food deprivation were monitored using bio-radar for a long time. It was found that the radar respiratory signals of animals in the decompensated period frequently changed from a sine curve to an “M” shaped curve. It was inferred that insufficient pulmonary surfactants caused by the rupture of lamellar bodies (LBs) in type II alveolar epithelial cells lead to respiratory distress syndrome. An “M” shaped curve is a feature of respiratory distress syndrome [16].

By summarizing the literature, we found that bio-radar can be used for the non-contact detection and localization of casualties under the conditions of penetrating non-metallic materials (such as bricks and wood). If more accurate physiological parameters of the injured can be extracted from the radar echo and used to determine the compression state of the injured, rescue workers can be guided to scientifically formulate rescue sequences and prepare relevant treatment drugs and medical equipment in advance to ensure the survival rate of the trapped injured person. However, one of the main challenges is that the vital signs of trapped individuals are easily affected by clutter and noise. Another challenge is identifying the compression state of injured individuals by fully utilizing hidden features in bio-radar data.

Currently, in the field of through-wall vital sign detection, commonly used bio-radars can be categorized into two types: continuous wave (CW) radar [24] and ultra-wideband (UWB) radar. CW radar applications are significantly limited, as they do not obtain distance information of the target. In contrast, UWB radar has attracted more and more attention due to its powerful penetrating capability and range-detecting capability. Among the sub-category of UWB radars, the frequency domain UWB radar and noise radar have disadvantages of more complex hardware implementation and the influence of sidelobes, respectively, even though they can achieve high receiver sensitivity. Therefore, mainly considering the penetration ability, range resolution ability of different targets at different range bins, and anti-interference ability, a time-domain UWB radar was applied for this study.

In summary, this study proposed a non-contact detection and recognition method based on bio-radar for the compression state of post-disaster wounds. It was divided into two parts: human-life signal extraction and post-disaster compression state recognition. The organizational structure of the study is as follows: In Section 2, a brief introduction is given to the working principle of UWB bio-radar and the preprocessing methods for raw radar echo data. In Section 3, for the extraction of human life signals, we used the PSO-VMD algorithm to filter out the clutter and noise in the echo signal and complete the reconstruction of human radar life signals based on permutation entropy to select effective modes. Section 4 showcases a post-disaster scenario simulated by designing experiments, and multiple non-contact life signals were collected from subjects in four typical post-disaster compression states to verify the reliability of the proposed method. Section 5 and 6 provides a discussion and conclusion, respectively. This study allows for the development and updating of casualty rescue strategies, thus reducing casualties in actual rescue scenarios [25,26].

## 2. UWB Bio-Radar

The UWB bio-radar is a pulse radio-based radar system that utilizes the Doppler effect to detect biological targets. Radar emits signals in the form of pulses with very short durations in the nanosecond range. The corresponding spectra of these emitted signals were concentrated in the frequency range of 30 MHz to 8000 MHz. The electromagnetic waves in these frequency bands have the ability to penetrate some non-metallic materials, making them suitable for application scenarios of penetrating debris and debris detection. Using a wide frequency band, the system can effectively obtain rich scattering information about the target, which is beneficial for the detection of detailed information.

When operating UWB bio-radar, the radio frequency (RF) synthesis unit generates a series of pulse signals with the same pulse repetition period under the control of a microcontroller in the radar transmitter. These pulse signals are split into two channels as trigger signals. One channel of the pulse generator generates an ultra-wideband signal, which is amplified by a power amplifier and transmitted into space. The other channel enters a delay circuit, which is adjusted by the microcontroller to generate a corresponding delay and form a local reference signal. The waveform of the echo reflected by the target is similar to that of the transmitted waveform but is loaded with the Doppler shift caused by the motion of the human surface. Subsequently, the antenna receives the modulated echoes, and the received signal was filtered and amplified using a low-noise amplifier. When these received signals reach the digital integrator inside the radar, there is a time shift between the echo and range-gate signals. If the time delay of the echo signal matches that of the range gate signal, a series of output levels are obtained. These levels are then sampled using an ADC and converted into digital signals, which are transmitted to an upper-level computer via a universal serial bus for further processing. The center frequency of the UWB bio-radar used in this study was 7.29 GHz, with a bandwidth of 1.4 GHz, a pulse repetition frequency of 15.18 MHz, and a slow-scanning frequency of 17 Hz. The radar system utilizes a pair of microstrip antennas placed side by side, which offer advantages such as small size, high gain, and good directivity. The fast time dimension contains 93 distance-sampling points, corresponding to a maximum detection distance of 5 m. The optimal detection angle is facing the human target, within an angle range of ±30 degrees in the sagittal plane. This deployment enables the radar to efficiently receive signals reflected from the body surface, enhancing the signal amplitude and signal-to-noise ratio. Once the angle range is exceeded, the target will almost completely out of the radar main lobe, resulting in a significant decrease in amplitude and SNR. The structure diagram of the UWB bio-radar system and the schematic diagram of non-contact cardiopulmonary activity detection are shown in Figure 1.

A fast-time signal is a data vector composed of all the distance sampling points of a bio-radar at a certain acquisition time. Fast-time signals at different times are sequentially concatenated to form a $D_{ori}\left(m\times n\right)$ data matrix. In this matrix, m=1,2,⋯,M represents the sampling points along the distance direction, which is the fast-time dimension, whereas n=1,2,⋯,N represents the sampling points along the data collection time, which is the slow-time dimension. Figure 2 shows the raw data matrix collected by the UWB bio-radar; Figure 2a shows a two-dimensional image based on the data matrix, and Figure 2b presents a three-dimensional image.

The collected raw radar echo data not only contain vital signs of the detected human body targets but also contain a large amount of static background interference. These static background interferences can be filtered out by subtracting the fast-time signal mean from the echo data. The background removal steps for the radar echo data can be expressed as
(1)$$D_{1}\left(m,n\right)=D_{ori}\left(m\times n\right)-\frac{1}{N}\sum_{n=1}^{N}D_{ori}\left(m\times n\right)$$

Additionally, electromagnetic attenuation occurs during the propagation of radar waves through obstacles, resulting in the received echo amplitude being significantly smaller than that of the initial transmitted pulse. If normalization processing is performed based on the collected channel signals, compensation can be provided for distant target echoes. The steps of attenuation compensation preprocessing can be expressed as
(2)$$D_{2}\left(m,n\right)=\frac{D_{1}\left(m,n\right)-\underset{1\leq l\leq L}{\min}\left[D_{1}\left(m,n\right)\right]}{\underset{1\leq l\leq L}{\max}\left[D_{1}\left(m,n\right)\right]-\underset{1\leq l\leq L}{\min}\left[D_{1}\left(m,n\right)\right]}$$

Finally, by calculating the energy sum of each position point in $D_{2}\left(m,n\right)$ along the fast-time dimension, the energy distribution curve $\left\{E\right\}=\left\{E_{1},E_{2},\cdots ,E_{L}\right\}$ for the fast-time dimension can be obtained. The maximum value of $E_{MAX}$ in the energy distribution curve can be searched to determine the fast-time sampling point at its corresponding position. The fast-time sampling point can be converted into the radial distance between the human target and the bio-radar detector. The results obtained after the preprocessing steps are shown in Figure 3, where Figure 3a represents the two-dimensional pseudocolor image after the preprocessing steps, and the results of the three-dimensional expansion are shown in Figure 3b. Using the radial distance calculation algorithm to determine the data vector corresponding to the fast-time position of the detection target in the radar echo data matrix and mapping it to the time domain, the radar echo signal at the human target position can be obtained, as shown in Figure 3c. It can be seen that the echo signal at this time already contains rich fluctuation information about the body’s surface, which means that the above preprocessing steps can eliminate the interference existing in the original radar echo signal.

## 3. Method

By preprocessing radar echoes, it is possible to suppress interference while extracting radar echo signals from the target location. However, because the vibrations caused by human heart motion on the body’s surface are much smaller than the chest expansion caused by breathing, the radar cardiac signals reflected in the echo signals are very weak and are easily overshadowed by strong background noise and slow-drift interference caused by slight movements of the human body. This makes it difficult to separate and reconstruct the radar vital signs of injured targets from the echoes, thereby affecting the determination of the compression state of injured individuals. To address these issues, this study proposed a non-contact sensing and identification method for the post-disaster compression state of injured individuals based on UWB bio-radar. The method consists of two steps: first, the reconstruction of radar vital signs from noisy echo signals, and second, the automatic recognition of the compression state of injured individuals using neural networks. A flowchart of the proposed method is shown in Figure 4.

### 3.1. Reconstruction of Human Radar Life Signals

#### 3.1.1. Principle of Variational Modal Decomposition

Variational mode decomposition (VMD) is an adaptive and completely non-recursive signal-processing method. The objective was to iteratively determine the optimal solution of the variational model and determine the bandwidth and center frequency of each mode. Therefore, the VMD can deconstruct a given radar echo $s\left(t\right)$ into K IMFs and ensure the estimated bandwidth and minimization of each IMF. The above process can be expressed as
(3)$$\underset{\left\{c_{i}\}\{w_{i}\right\}}{\min}\left\{{\left.\sum_{i}\left\|\partial_{t}\left[\left(\delta \left(t\right)+\frac{j}{\pi t}\right)+c_{i}\left(t\right)\right]\right.e^{-jw_{i}t}\right\|}_{2}^{2}\right\}$$

$c_{i}\left(t\right)$ represents the *i*-th IMF obtained through decomposition, and $\omega_{i}$ is the center frequency corresponding to the *i*-th IMF. By introducing the quadratic penalty term α and Lagrange multiplier $\lambda \left(t\right)$, the constraint problem is transformed into a non-constraint problem:(4)$$\begin{array}{l} L\left(\{c_{i}\left(t\right)\},\{\omega_{i}\},\lambda \left(t\right)\right)\\ =\alpha {\left.\sum_{i}\left\|\partial_{t}\left[\left(\delta \left(t\right)+\frac{j}{\pi t}\right)\times c_{i}\left(t\right)\right]\right.e^{-j\omega_{i}t}\right\|}_{2}^{2}+{\left.\left\|f\left(t\right)-\sum_{i}c_{i}\left(t\right)\right.\right\|}_{2}^{2}+\langle \lambda \left(t\right),f\left(t\right)-\sum_{i}c_{i}\left(t\right)\rangle \end{array}$$

Subsequently, the variational problem of the VMD method is solved using the alternating direction multiplier iteration algorithm, and ${\hat{c}}_{i}^{n+1}\left(t\right)$, ${\hat{\omega }}_{i}^{n+1}$, and ${\hat{\lambda }}^{n+1}\left(t\right)$ are iteratively updated using Equations (5)–(7):(5)$${\hat{c}}_{i}^{n+1}\left(\omega \right)=\frac{\hat{f}\left(\omega \right)-\sum_{i\neq k}{\hat{c}}_{i}\left(\omega \right)+\frac{\hat{\lambda }\left(\omega \right)}{2}}{1+2\alpha {\left(\omega -\omega_{i}\right)}^{2}}$$
(6)$${\hat{\omega }}_{i}^{n+1}=\frac{\int_{0}^{\infty }\omega {\left|{\hat{c}}_{i}\left(\omega \right)\right|}^{2}d\omega }{\int_{0}^{\infty }{\left|{\hat{c}}_{i}\left(\omega \right)\right|}^{2}d\omega }$$
(7)$${\hat{\lambda }}^{n+1}\left(\omega \right)={\hat{\lambda }}^{n}\left(\omega \right)+\tau \left(\hat{f}\left(\omega \right)-\sum_{i}{\hat{c}}_{i}^{n+1}\left(\omega \right)\right)$$

Finally, given the accuracy ε, until the condition $\sum_{i}\left[\left\|{\hat{c}}_{{}_{i}}^{n+1}\left(t\right)-{\left.{\hat{c}}_{{}_{i}}^{n}\left(t\right)\right\|}_{2}^{2}\right./\left\|{\left.{\hat{c}}_{{}_{i}}^{n}\left(t\right)\right\|}_{2}^{2}\right.\right]<\varepsilon$ is met, an exit needs to happen. Otherwise, the above iterative steps will continue to be performed to ultimately obtain the optimal solution to the unconstrained problem. On the one hand, VMD utilizes its inherent Wiener filtering characteristics to achieve better noise-filtering performance compared with algorithms, such as EMD, and on the other hand, due to its lack of adaptive characteristics, such as EMD and LMD, the effectiveness of VMD largely depends on the user’s selection of the quadratic penalty term α and the number of submodes K for VMD. If the K value is too small, it will cause aliasing, while if it is too large, it will lead to excessive signal decomposition. In addition, if the value of α is too large, it will cause the frequency bands of each mode to be too narrow, resulting in the loss of useful information. In contrast, it will cause the frequency band to be too wide and carry excess interference information in the mode. Therefore, the user experience needs to be relied on to manually adjust the parameters α and K, which makes it difficult to achieve the optimal decomposition effect.

#### 3.1.2. Particle Swarm Optimization for VMD

Particle swarm optimization (PSO) is an evolutionary computation technology. It adjusts the fitness of particles to the environment by iteratively comparing the optimal state of individual particles and the global optimal state in the fitness function area and then finds the optimal solution in the area. When tuning the parameters of VMD, selecting the appropriate fitness function is the key to the smooth operation of PSO. Due to the influence of the parameters α and K, each IMF decomposed using VMD often exhibits varying degrees of artifacts. The strengths of these artifacts can be evaluated by comparing the number of interrelationships between each IMF and the original signal and then quantifying them by calculating the overall mean of these correlation coefficients. Taking the correlation number $R_{i}$ between the *i*-th IMF and the original signal as an example, the formula is as follows:(8)$$R_{i}=\frac{\sum_{t=1}^{T}\left(s\left(t\right)-\bar{s}\right)\left(c_{i}\left(t\right)-\bar{c}\right)}{\sqrt{\sum_{t=1}^{T}{\left(s\left(t\right)-\bar{s}\right)}^{2}\sum_{t=1}^{T}{\left(c_{i}\left(t\right)-\bar{c}\right)}^{2}}}$$

$s\left(t\right)$ is the preprocessed radar echo signal, $c_{i}\left(t\right)$ is the *i*-th IMF, and T represents the sampling duration; the mean correlation number between each IMF and the original signal can be expressed as $\bar{R}=\sum_{i=1}^{n}R_{i}/n$. The variance index between the correlation numbers is introduced to eliminate fluctuations in the mean value of the correlation numbers. Finally, a scheme that uses the maximum ratio of the mean value of the correlation numbers and its variance as the fitness function was formulated to optimize the VMD parameters. Maximum cross-correlation (MCC) is introduced as the optimality function in PSO and is expressed as follows:(9)$$F=\frac{\bar{R}}{\sum_{i=1}^{n}{\left(R_{i}-\bar{R}\right)}^{2}/n-1}$$

n represents the number of decomposed modes and $\bar{R}$ represents the mean of the cross-correlation coefficient. After the PSO parameter initialization, the parameters [α,K] of the VMD were iteratively optimized within the ranges [200, 2000] and [2, 8], respectively. Figure 5 shows the results of the fitness function for different parameter combinations. The red area in the figure indicates that when α=1053 and K=4, it will minimize the artifact components contained in the decomposed IMF, which means that they have the strongest correlation with the original signal.

#### 3.1.3. Life Signal Reconstruction Based on Permutation Entropy

Owing to the sensitivity of VMD to noise, the partial IMF center frequency $\omega_{k}$ falls within the main noise frequency band. Permutation entropy (PE) is a method used to detect the randomness of a time series and quantitatively evaluate the random noise contained in signal sequences. Therefore, the degree of noise in each mode can be quantitatively evaluated using the permutation entropy. High-noise modes were automatically identified and removed according to the set entropy threshold. A time series $X\left(i\right)$ of length N was phase-space-reconstructed and the reconstruction matrix Y was obtained:(10)$$Y=\left[\begin{array}{cccc} x\left(1\right) & x\left(1+\tau \right) & \cdots & x\left(1+\left(m-1\right)\tau \right)\\ x\left(2\right) & x\left(2+\tau \right) & \cdots & x\left(2+\left(m-1\right)\tau \right)\\ \begin{array}{l} x\left(j\right) \end{array} & \begin{array}{l} x\left(j+\tau \right) \end{array} & \begin{array}{l} \cdots \\ \vdots \end{array} & \begin{array}{l} x\left(j+(m-1)\tau \right) \end{array}\\ x\left(K\right) & x\left(K+\tau \right) & \cdots & x\left(K+\left(m-1\right)\tau \right) \end{array}\right]$$

m represents the embedding dimension, τ represents delay time, and $K=N-\left(m-1\right)\tau$. Each row in the matrix can be regarded as a reconstruction component, and Y has a total of K reconstruction components. The *j*-th reconstruction component $\left(x\left(j\right),x\left(j+\tau \right),\cdots ,x\left(j+(m-1)\tau \right)\right)$ is reconstructed in the $X\left(i\right)$ reconstruction matrix in ascending order based on the numerical value, and $j_{1},j_{2},\cdots ,j_{m}$ represents the column index where each element in the reconstruction component is located:(11)$$x\left[i+\left(j_{1}-1\right)\tau \right]\cdots x\left[i+\left(j_{2}-1\right)\tau \right]\leq \cdots \leq x\left[i+\left(j_{m}-1\right)\tau \right]$$

Therefore, the column indexes form a set of symbol sequences:(12)$$S\left(l\right)=\left(j_{1},j_{2},\cdots ,j_{m}\right)$$

l=1,2,⋯,k, k≤m!, and there are a total of m! types of symbol sequences of m-dimensional phase space mapping. Subsequently, the probability of occurrence of each symbol sequence $P_{1},P_{2},\cdots ,P_{k}$ is calculated by dividing the number of occurrences of the symbol sequence by m!. Then, the entropy of k different symbol sequences in the time series $X\left(i\right)$ can be defined as
(13)$$H_{pe}=-\sum_{j=1}^{k}P_{j}\ln\left(P_{j}\right)$$

Using $H_{pe}=H_{pe}/\ln\left(m!\right)$ normalizes the entropy value to the interval of [0, 1]. The value of the entropy represents the randomness of the time series $X\left(i\right)$. Choosing an appropriate permutation entropy threshold is the key to automatically removing noise, and is the foundation to ensure accurate life signals reconstruction. After several testing experiments, it was found that the permutation entropy threshold of 0.78~0.8 was more suitable according to the linear relationship between the randomness of the time series and its corresponding $H_{threshold}$ value. In this situation, IMFs above $H_{threshold}$ can be removed. Finally, reconstructing the target life signal based on the preserved IMF effectively preserves the useful human life signals while removing noise. The specific implementation steps are as follows:(1)The maximum cross-correlation between each IMF and the radar echo signal after VMD processing was used as the fitness function of the PSO to optimize the VMD parameters.(2)The radar echo signal was processed based on the parameter-optimized VMD and decomposed into a series of IMFs;(3)The entropy value of each IMF arrangement was calculated separately;(4)The entropy values of each IMF were compared and filtered with $H_{threshold}=0.8$, high-noise IMFs larger than $H_{threshold}$ were directly removed, and the remaining IMFs were reconstructed to restore the radar life signal of the human target. A flowchart of this method is shown in Figure 6.

### 3.2. Identification of the Compression State of the Injured

#### 3.2.1. Building the Radar Life Signal Time-Frequency Dataset

Performing time-frequency domain analysis on the reconstructed human radar life signal can obtain the corresponding time-frequency images. These images can intuitively reflect the variation in the frequency components with time. Compared with one-dimensional radar life signals, such as breathing and a heartbeat, the time-frequency images are more suitable as inputs for network models to achieve identification of the compression state. A continuous wavelet transform uses finite length and convertible wavelet basis functions to enhance its local generalization ability. The commonly used wavelet bases include Haar and Morlet wavelets, Symlet, and the Db wavelet series. In order to select wavelet basis functions suitable for analyzing radar life signals, the four wavelet bases mentioned above were used individually to process the reconstructed radar life signals. The results based on different wavelet bases are shown in Figure 7.

It can be seen that the Morlet wavelet had a clearer time-frequency resolution and was selected for the time-frequency analysis of the reconstructed radar life signals. The processed time-frequency images were used as the dataset for the subsequent network training and recognition.

#### 3.2.2. Establishment of the Compression State Identifying Network

As a hierarchical neural network, a convolutional neural network (CNN) essentially imitates the working principles of the human visual system by constructing convolutional kernels of different sizes to extract unique local features from input data, thereby achieving accurate data recognition. A large number of experiments [27] showed that when performing complex classification tasks, the deeper the network structure can learn and capture different features from the data, the stronger the ability to express data features. However, in practical applications, increasing the depth of a neural network increases the number of weight parameters to be updated, making the network increasingly difficult to train.

To prevent gradient dispersion and network degradation caused by deepening layers while fully extracting feature information, this study proposed an improved CNN scheme based on multiscale feature extraction and a residual structure by referring to the inception structure in the GoogLeNet neural framework and combining it with residual networks. First, in a multiscale feature-extraction structure, parallel stacking is used to increase the network width, and the feature-extraction ability is improved by utilizing convolutional kernels of different sizes. The feature dimensions of the different branches were then concatenated to obtain a set of multichannel feature maps. For feature maps with a large number of channels, a channel attention mechanism is used to help the network learn more important feature channels. This not only improves the expression and classification performance of the network but also reduces the impact of redundant information and improves the robustness and generalization ability of the network. Finally, a 1 × 1-pixel convolutional kernel in the inception structure is used to downscale the input feature map, achieving the goal of reducing the amount of parameter computation, and thus, accelerating the network model training.

Additionally, in a traditional CNN, the convolutional modules in the upper layer and the convolutional modules in the lower layer are usually end-to-end, but such a network structure often prevents the latter from utilizing the initial input information of the previous convolutional module, thereby limiting the feature-learning ability of the convolutional layer on the input data. Therefore, by introducing a residual structure with the characteristic of “cross layer connection”, the network can synchronously transmit the original features to the next layer during forward propagation, thereby avoiding information loss or redundancy. Finally, the extracted features are converted into one-dimensional data through the fully connected layer, and the classification results of the human compression state are output through the cross-entropy loss function. The structure of the multiscale feature learning network is shown in Figure 8.

The detailed network parameters are shown in Table 1.

In order to recognize the compression state of the injured, it is necessary to preprocess the raw echo radar data first, and then use the PSO-VMD algorithm based on permutation entropy to denoise the preprocessed signal and reconstruct the radar human life signal. Due to the fact that the human vital sign characteristic information is mainly contained in the spectrum of the reconstructed radar life signal, the reconstructed human life signal is conducted via time-frequency analysis. Then, the time-frequency maps with compression state information are sent into a network to complete the training process. Finally, the trained network is applied to evaluate the type of compression state. Figure 9 shows the flow chart of the proposed recognition method.

## 4. Experiment Results

### 4.1. Experiment on Reconstructing Human Life Signals

#### 4.1.1. Indoor Free Space Detection Experiment

An indoor free-space detection experiment was conducted to verify the performance of the proposed PSO-VMD algorithm based on permutation entropy. The subjects were detected without any obstacles by using UWB bio-radar. Simultaneously, two contact detection devices—an RM6240E pressure breathing detector and an IX-B3G portable electrocardiograph—were used for synchronous data collection as a reference. The proposed algorithm was applied to extract radar respiratory and cardiac signals from the radar echo signals. Finally, the radar respiratory and cardiac signals were compared with contact reference signals. Figure 10 shows the experimental setup under indoor free-space detection conditions.

Figure 11 shows the results of the contact and non-contact data from the same subject in an indoor free-space detection scenario. The black solid line represents the contact reference signal, and the red dashed line represents the processed radar signals.

The signals collected by the contact and contactless devices exhibited strong consistency. As shown in Figure 11b,d, after a fast Fourier transform, the main frequencies of the radar respiratory signal and radar heartbeat signal were 0.41 Hz and 1.2 Hz, respectively.

#### 4.1.2. Outdoor Obstacle Barrier Scenario Detection Experiment

It should be noted that some damp building materials will cause significant losses above 1 GHz. In this work, to mainly verify the compression states recognizing the effectiveness of the proposed method, the optimal test scenario with an obstacle barrier of multi-layer dry gypsum board was set. Figure 12 illustrates the experimental setup under outdoor-obstructed detection conditions.

In this experiment, five signals were collected from each participant, each lasting 30 s. Using the contact signal as a reference, the average respiratory rate and heart rate measurement error rates of each subject for the five signals are shown in Figure 13.

Figure 13 shows that the average respiratory rate measurement error rate of all subjects was 2.53%, and the average heart rate measurement error rate was 5.24%, which met the accuracy requirements for extracting injury target life signals in post-disaster rescue scenarios. The effectiveness of the proposed PSO-VMD based on permutation entropy was demonstrated.

### 4.2. Compression State Identification Experiment

To replicate a scene of human bodies trapped under the ruins of squeezing, radar signal collection experiments were conducted on 10 adult subjects (six males and four females) aged between 21 and 45 years using a self-developed squeezing platform. All ten volunteers signed an informed consent form. A self-developed squeezing platform was designed to simulate heavy objects pressing on a human torso after a disaster. To better replicate a real post-disaster scene and ensure the subjects’ safety simultaneously, the maximum weight that the subjects could accept was used as the upper weight limit of the heavy objects applied in the experiment, and two different states, namely, “compression” and “no compression”, were set. Considering the different trapped postures of buried personnel after actual disasters, two different postures, namely, “supine” and “prone”, were also set in this experiment. The four types of states for subjects with different compression levels and postures are listed in Table 2.

In the “no compression” state, four layers of 15 mm thick gypsum board were used as the obstacle between the subject and the radar, and the obstacle did not contact the subject’s body. The “compression” state refers to the situation where bricks were placed on a wooden board to build a compressive device with a total weight of 10 kg. The device was pressed on the subject’s body with its weight regardless of whether the subject was in a supine or prone position. This pressure was evenly distributed throughout the torso of the subject. The intensity of the pressure was determined by the contact area between the compressive device and the human body. Therefore, the pressure and intensity of pressure were independent of the subject’s weight, gender, and posture. To expand the total number of signals, four random detection distances between the UWB bio-radar and the subject within the range of 1.5 to 2 m were set to collect signals for each subject (40 signals for 10 people). Four sets of signals with different compression levels and postures were collected at each distance with a single collection time of 180 s. A control group under the detection scene with no target was also set up to simulate an actual post-disaster rescue. Figure 14 shows the four types of experimental scenarios with subjects under different compression states.

To avoid overfitting owing to the small sample size of the training set, a data augmentation method was adopted to expand the training set sample size. By adding random noise to the time-frequency images, the dataset was expanded three times to the original size, and the training and test sets were configured in a ratio of 8:2. This not only expanded the number of training sets but also enabled the neural network to adaptively learn to improve its anti-noise ability during the training process, which further enhanced the anti-noise performance and classification generalization ability. Table 3 provides a detailed list of the dataset compositions.

This study used Tensorflow 2.0 to build a classification network for compression state recognition and trained it based on the above dataset. The loss and accuracy curves in the training phase are shown in Figure 15, while Figure 16 shows the confusion matrix results of the compression state recognition for the test set.

The overall classification accuracy of the network and the recall and precision of each category were calculated and are listed in Table 4. The overall classification accuracy and macro-F1 values of the model were 0.9278 and 0.9422, respectively. By observing the results of the confusion matrix, it was found that the network accurately identified the presence of a target in the detection area and distinguished the trapped postures. However, the recall rate for the category of “Under compression in supine state” was only 0.726. According to the confusion matrix, 27% of samples were incorrectly identified as being under compression in the prone position. The reason for this phenomenon may be that as the human body is compressed for a longer time, the breathing mode of the subject in the supine compressed state may gradually converge to the prone posture owing to muscle paralysis and respiratory distress. Although the recall rate of the category of “under compression in supine state” was relatively low, the network did not classify the subject as the wrong type of pressure. Therefore, the network met the correct recognition rate requirements for the status of squeezed subjects during a post-disaster rescue.

To further verify the superiority of this method, the network was compared with a support vector machine (SVM) based on respiratory features and a 1D-CNN network based on radar life signals. The SVM based on respiratory features was shown to exhibit good classification performance in the field of human and animal identification using bio-radar [15]. To successfully apply an SVM to compression state recognition, it is necessary to extract and analyze features from radar life signals. In this study, a total of 10 respiratory features containing the average respiratory amplitude, linear peak deviation of respiration, average velocity of expiration and inspiration, and average interval time between expiration and inspiration were selected for the dataset production and the training of the SVM. The 1D-CNN has a network structure similar to that of a traditional CNN. Except for the input data being a one-dimensional time series, the main difference is that the one-dimensional array in the hidden layer replaces the two-dimensional matrix to handle kernel functions and feature maps. This network had four convolutional layers, four pooling layers, and a fully connected flattening layer. The processed radar life signal was transformed into a series of feature vectors through the first convolutional module, and then downsampling was completed through the pooling layer. The above steps were repeated until the end, and the output feature map of the previous layer was sent to the fully connected flattening layer and transmitted to a softmax function to complete the five classifications of the compression state. In the comparison experiment, the original dataset was divided randomly into three parts: A, B, and C. During training, one of the three datasets was used as the training dataset and the other two parts were selected with one of them as the test dataset. These three methods were used to identify the state of the compression victims. The experimental results are shown in Figure 17. A-B and A-C in the figure represent the use of dataset A as the training sample and datasets B and C as test samples, respectively.

From Figure 17, it can be observed that the accuracy of the proposed method was higher than that of the other two methods. The average recognition accuracy of the CNN based on multiscale feature extraction and residual structure was 92.59%, whereas the average recognition accuracies of the SVM based on respiratory features and 1D-CNN based on radar life signals were 88.48% and 86.58%, respectively. In summary, the proposed method could fully extract the vital sign information contained in the radar life signal while avoiding degradation caused by the deepening of the network layers. Compared with the SVM and 1D-CNN, the recognition accuracies of the proposed method improved by 4.19% and 6.01%, respectively.

Owing to the correlation between the probability of a subject suffering from crush syndrome and the duration of compression, the longer the duration of compression, the more severe the crush injury and the higher the incidence of crush syndrome. Therefore, wounded patients who may suffer from crush syndrome should be identified as soon as possible and the crushing of heavy objects should be relieved as soon as possible. If the injured limbs are exposed, a tourniquet should be used at the proximal end of the injured limb, and an alkaline liquid should be injected to correct acidosis and hyperkalemia and prevent the occurrence of acute renal failure.

## 5. Discussion

This study focused on the various challenges faced by bio-radar in the field of post-disaster life detection and proposed a full-process solution based on UWB bio-radar for post-disaster casualty life signal detection and compression state recognition. In this scheme, a parameter-optimized VMD was proposed to process the radar echo signal, and the permutation entropy algorithm was applied to achieve a fast and accurate reconstruction of the subject life signal. Subsequently, an improved CNN based on multiscale feature extraction and a residual structure was proposed to achieve automatic recognition of the compression state. This network could fully learn the input data feature information while preventing gradient dispersion and network degradation caused by deepening the layers. According to the experimental results, the accuracy of this method in identifying the compression state of the subject improved by 4.19% and 6.01% compared with the SVM and 1D-CNN, respectively. This study is of great significance for extracting the vital signs of squeezed subjects and guiding rescue workers in performing correct and scientific rescues in practical applications.

However, this study has certain limitations. On the one hand, the trapped postures of the simulated wounded were only summarized as supine and prone in this work, without fully considering the various trapped postures that a rescue worker may face after a disaster occurs. To protect the subjects outdoors during the obstacle experiment, we controlled the weight of the cover plate acting on the human torso to only 10 kg. In actual disasters, the weight of the crushed ruins may be much greater than this value. In addition, the rubble that covers the injured in a real-life earthquake is randomly orientated and highly scattered. Therefore, in the future, we will construct more realistic post-disaster rescue scenarios and apply different modulation schemes of UWB radar to research more complex and heavier compression state identification.

## 6. Conclusions

Owing to its ability to detect vital signs caused by the human heart and lung activity at a certain distance and through certain obstacles, UWB bio-radar is often used in post-disaster searches and rescues to detect the life signals of subjects in buried situations. However, noise from the detection environment and spontaneous sway interference of the trapped human body cause the detected radar life signals to exhibit a low signal-to-noise ratio and strong randomness, making it difficult to obtain more effective information and further evaluate the compression state. Focusing on the above problems, we proposed a method that combines PSO-VMD based on permutation entropy and an improved CNN based on multiscale feature extraction and residual structure. The effectiveness and progressiveness of this method were verified with experimental results. This study provides a foundation for further in-depth analyses of the life status of trapped individuals. This novel method is expected to be applied in post-disaster rescues, battlefield searches, and rescue tasks and can provide guidance for the optimization of rescue strategies and rational allocation of medical resources, promote the safety of trapped personnel, and reduce the casualty rate in disaster relief areas.

## Figures and Tables

**Figure 1 bioengineering-10-00905-f001:**
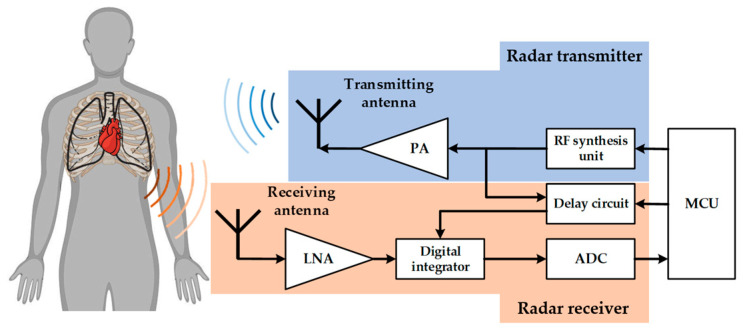
Structure diagram of a UWB bio-radar system and schematic diagram of non-contact cardiopulmonary activity detection.

**Figure 2 bioengineering-10-00905-f002:**
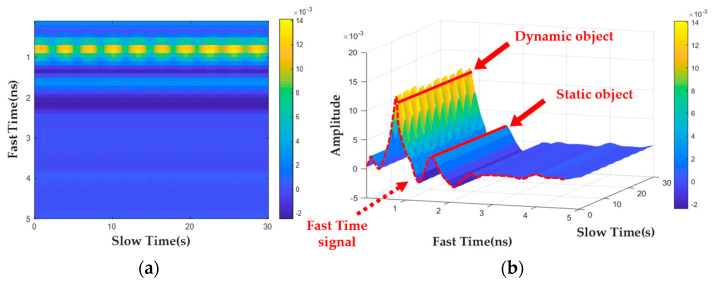
Original UWB radar data map: (**a**) two−dimensional color map; (**b**) three−dimensional color map.

**Figure 3 bioengineering-10-00905-f003:**
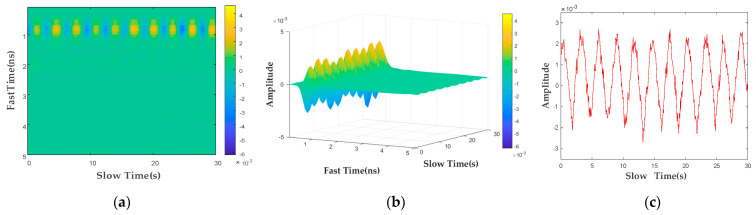
Preprocessed UWB radar data map: (**a**) two−dimensional color map; (**b**) three−dimensional color map; (**c**) radar echo signal at a human body’s position.

**Figure 4 bioengineering-10-00905-f004:**
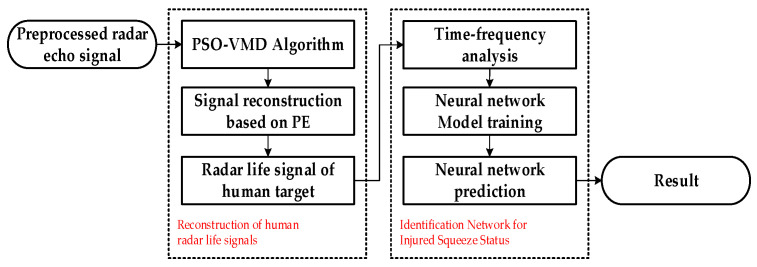
Overall process block diagram.

**Figure 5 bioengineering-10-00905-f005:**
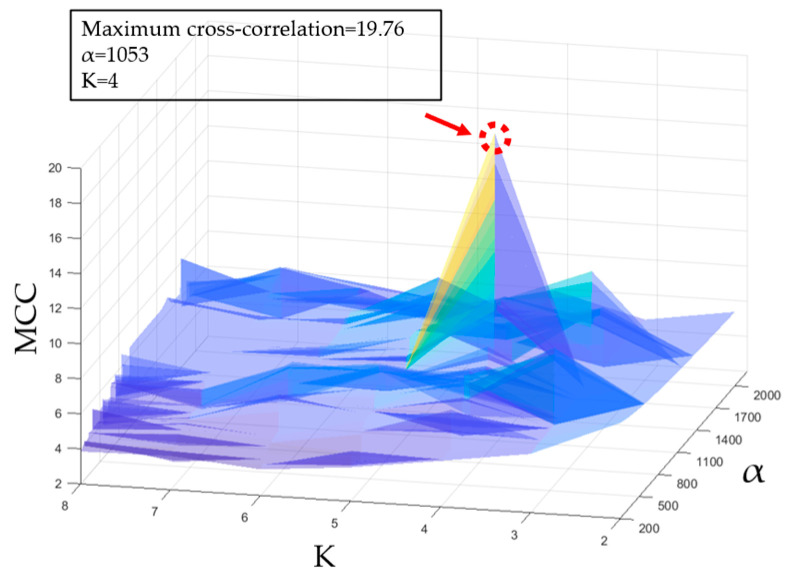
PSO diagram: the distributions of the MCC under different combinations of α and K. The optimization parameters: MEE = 19.76, α = 2062, and K = 4.

**Figure 6 bioengineering-10-00905-f006:**
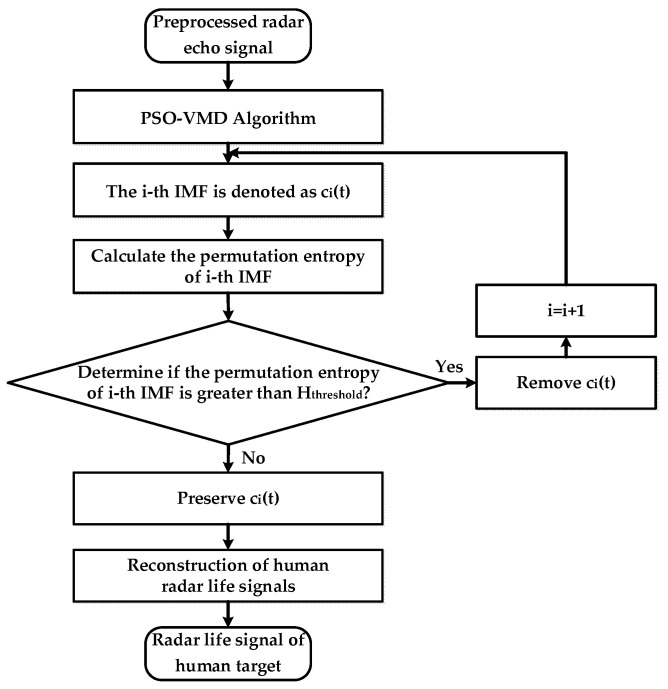
Algorithm flow chart of PSO-VMD based on permutation entropy.

**Figure 7 bioengineering-10-00905-f007:**
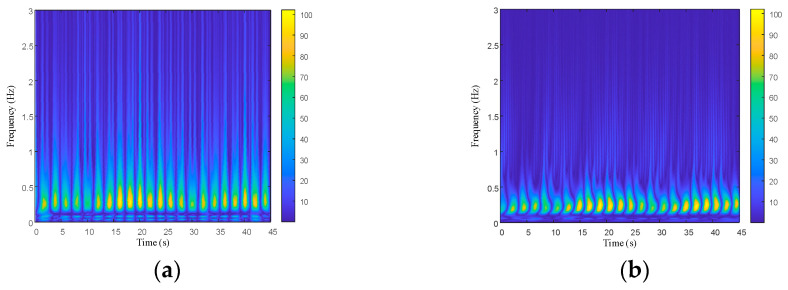
Wavelet time-frequency diagram of the simulated signal: (**a**) Haar wavelet; (**b**) Db4 wavelet; (**c**) Sym4 wavelet; (**d**) Morlet wavelet.

**Figure 8 bioengineering-10-00905-f008:**
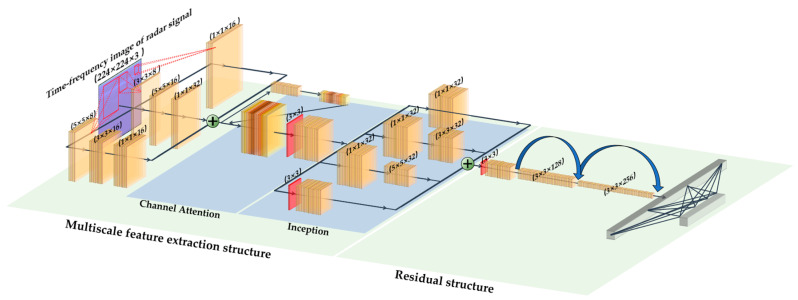
Compression state recognition network structure.

**Figure 9 bioengineering-10-00905-f009:**
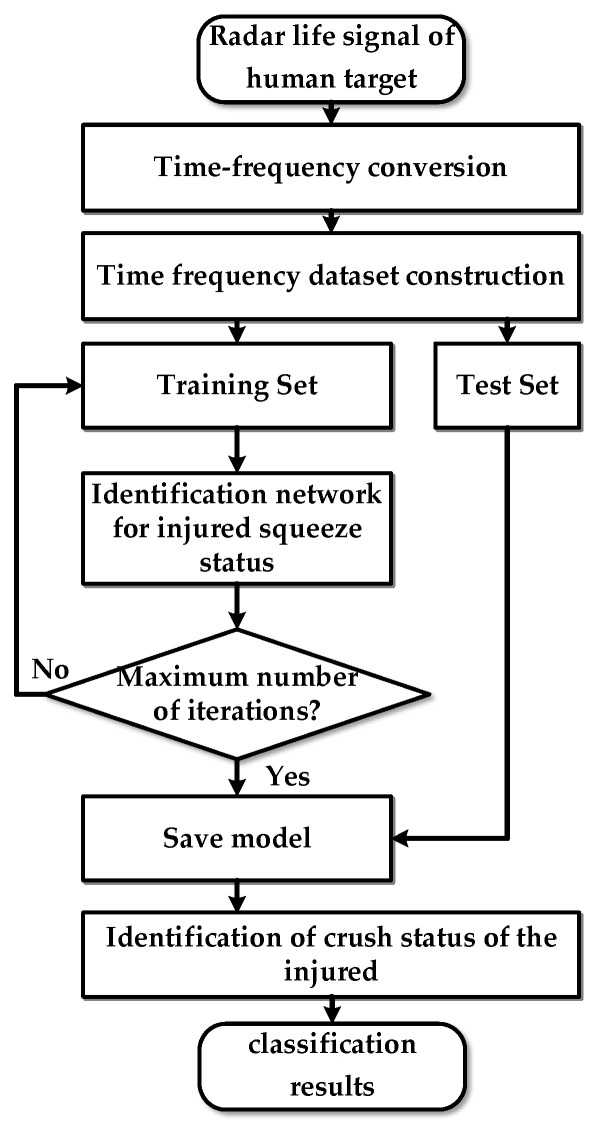
Flow chart of the human compression state recognition process.

**Figure 10 bioengineering-10-00905-f010:**
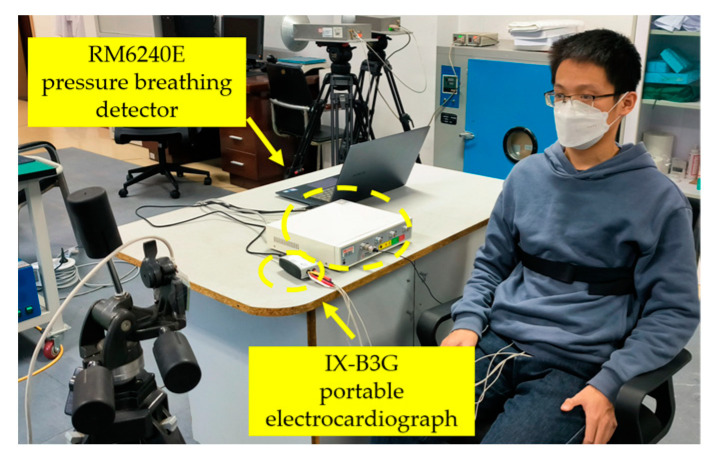
Experimental scenarios for extracting contact and non-contact human life signals under indoor free space conditions.

**Figure 11 bioengineering-10-00905-f011:**
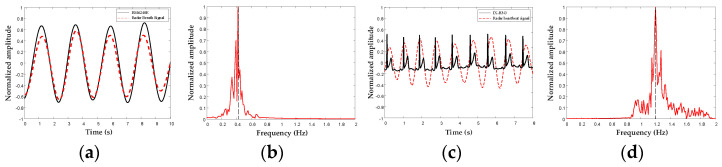
Schematic diagram of comparison between radar life and reference signals: (**a**) time domain comparison of respiratory signals; (**b**) radar respiratory signal spectrum; (**c**) time domain comparison of cardiac signals; (**d**) radar heartbeat signal spectrum.

**Figure 12 bioengineering-10-00905-f012:**
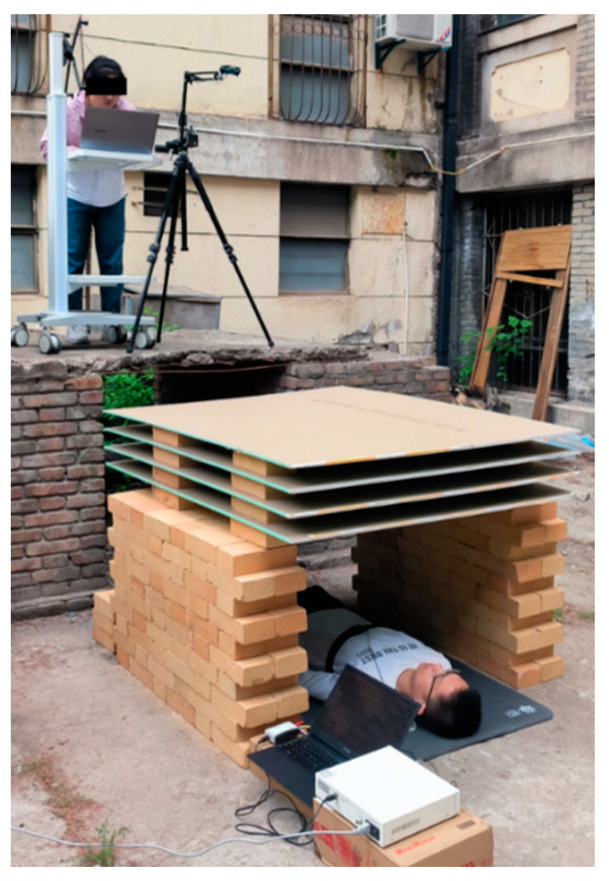
Experimental outdoor obstructed detection scenario for extracting contact and non-contact human life signals.

**Figure 13 bioengineering-10-00905-f013:**
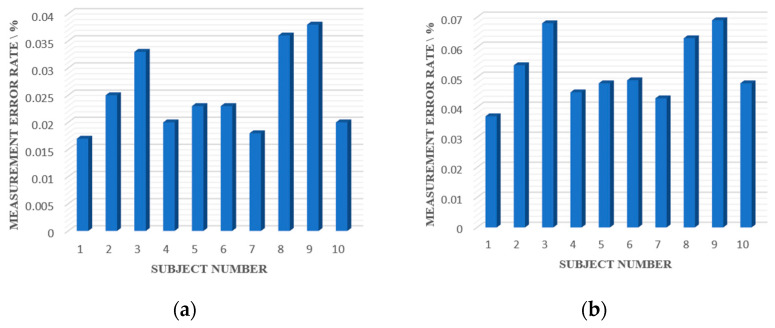
Average error rate between the radar life signals versus reference signals: (**a**) respiratory rate measurement error rate; (**b**) heart rate measurement error rate.

**Figure 14 bioengineering-10-00905-f014:**
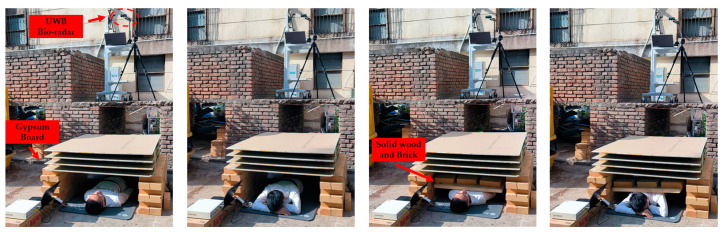
Four types of compression state identification experimental scenarios.

**Figure 15 bioengineering-10-00905-f015:**
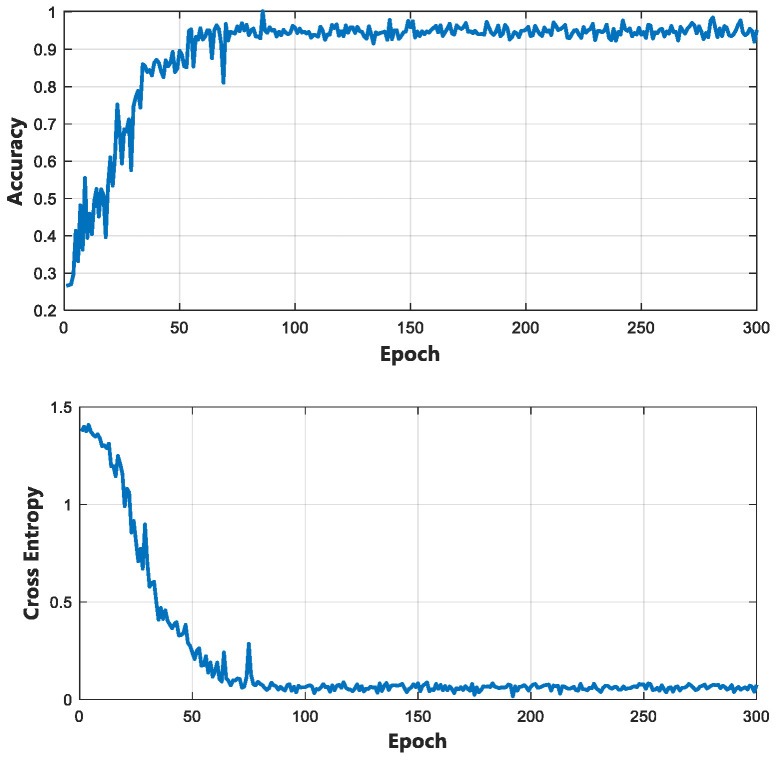
Accuracy and loss curves of the network training process.

**Figure 16 bioengineering-10-00905-f016:**
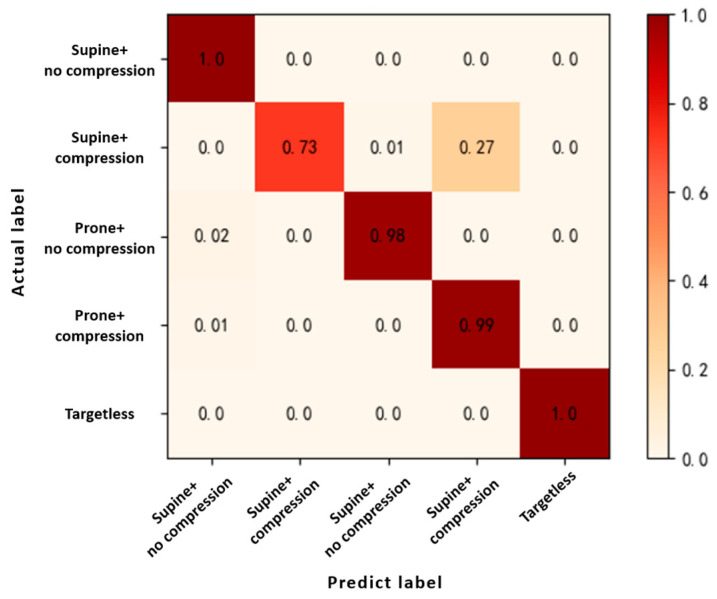
Confusion matrix of the compression state recognition.

**Figure 17 bioengineering-10-00905-f017:**
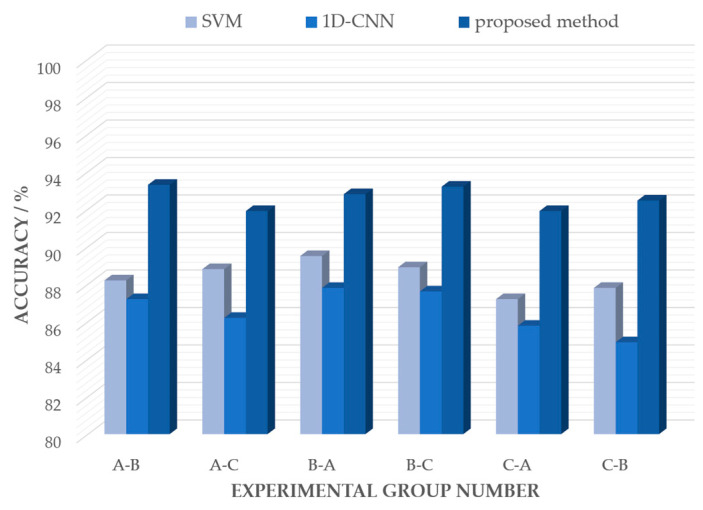
Multi-model cross-validation results.

**Table 1 bioengineering-10-00905-t001:** Network parameters.

Network Module	Hierarchy	Network Parameters			
		Number of Input Channels	Number of Output Channels	Kernel Size	Step
Multiscale feature extraction	Convolution	3	16	(1 × 1)	1 × 1
	Convolution	3	32	(3 × 3) (5 × 5) (1 × 1)	1 × 1
	Convolution	3	16	(5 × 5) (3 × 3) (1 × 1)	1 × 1
Channel attention mechanism	Pooling	~	~	(3 × 3)	2 × 2
Inception-Resnet	Convolution	32	32	(1 × 1)	1 × 1
	Convolution	32	32	(1 × 1) (3 × 3)	1 × 1
	Convolution	32	32	(1 × 1) (5 × 5)	1 × 1
	Pooling	~	~	(3 × 3) (1 × 1)	2 × 2
	Convolution	32	128	(3 × 3)	1 × 1
	Convolution	128	256	(3 × 3)	1 × 1
	Pooling	~	~	(3 × 3)	2 × 2

Note: “~” indicates no parameters.

**Table 2 bioengineering-10-00905-t002:** Four typical compression states.

	Compression	No Compression
Supine	Under compression in a supine state	Not compressed in a supine state
Prone	Under compression in a prone position	Not compressed in a prone position

**Table 3 bioengineering-10-00905-t003:** Dataset composition.

Squeeze State Type	Number of Training Set Samples	Number of Test Set Samples
Supine + no compression	720	180
Supine + compression	720	180
Prone + no compression	720	180
Prone + compression	720	180
Targetless control group	240	60
total	3120	780

**Table 4 bioengineering-10-00905-t004:** Macro-F1 score, accuracy, recall, and precision.

	Control Group (No-Target)	Supine		Prone	
		No Compression	Compression	No Compression	Compression
Recall	1.000	1.000	0.726	0.977	0.988
Precision	1.000	0.966	1.000	0.994	0.773
Accuracy	0.9278				
Macro-F1	0.9422				

## Data Availability

Due to the involvement of multiple collaborating institutions in this study, we have decided not to disclose the dataset temporarily after thorough discussion.

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
