# Peer review of "A Novel Non-Contact Detection and Identification Method for the Post-Disaster Compression State of Injured Individuals Using UWB Bio-Radar"

_bioengineering, 2023, doi:10.3390/bioengineering10080905_

Round 1
Reviewer 1 Report
The proposed technique is quite complex while in comparison to the techniques present in the literature.
It is not clear which sort of antennas are used in the transmitter and receiver side.
How the antennas will be mounted whether it will be placed side by side or face to face. What would be the impact by placing at any angular position?
It is very difficult to create a delay circuit at IR-UWB while looking from time domain perspective.
The are multiple noise in the output results.
N/A
Reviewer 2 Report
This technology has been studied in detail by various researchers in different countries since the 80s. However, it did not find any noticeable success in its practical applications. This work represents another attempt to make progress in this direction.
In general, the work is well written and is of interest to researchers working in this field.
The paper provides a detailed overview of works on this subject, but one might get the impression that only impulse radars are used for this purposes, although radars with a continuous signal have also been studied for a long time, see for example:
A. S. Bugaev, V. V. Chapursky, S. I. Ivashov, V. V. Razevig, A. P. Sheyko and I. A. Vasilyev, "Through wall sensing of human breathing and heart beating by monochromatic radar," Proceedings of the Tenth International Conference on Grounds Penetrating Radar, 2004. GPR 2004., Delft, Netherlands, 2004, pp. 291-294.
I recommend the authors to correct this shortcoming.
In general, the work can be published without significant changes.
Reviewer 3 Report
The title [IR-UWB Bio-Radar] of the paper is quite confusing because its not a radio and IR can be confused with Infra Red. The title needs to the changed to UWB Bio Radar and if the authors wish to distinguish further, then UWB [TD] where TD stands for Time Domain
The authors state "The electromagnetic waves in these frequency
bands have strong penetration capabilities through nonmetallic
materials" This statement needs to be qualified because at 7.29 GHz with a bandwidth of 1.4 GHz the loss in dB per metre of many materials such as concrete, bricks or natural material may be very high and losses in the region of 100's of dB/m can be expected, so the statement is not really credible.
There is no quantification of the loop gain of the radar nor its minimum detectable signal level and this a a major weakness of the paper in that readers will have difficulty in understanding the fundamental performance of the radar. What is the effective radar cross section of the targets and when do the authors expect that targets will be become undetectable as a function of range, body mass and intervening materials? A graph might be helpful
The test results in section 4 do not provide information on the losses of the multiple layers, so the reader has no idea what is the real capability of the radar and its processing. The authors need to provide this infomation.
The compressive state is not well described. What does this mean in terms of the peak pressure applied to the subjects. Is this the distributed pressure over the body area or the peak pressure on say a small section of the rib cage of the male / female subjects and in relation to their individual body mass? The authors need to provide better information on this aspect.
The title A Novel Non-Contact Detection and Identification Method for Post-Disaster Compression State of Injured Individuals Using IR-UWB Bio-Radar of the paper is quite confusing, because its not a radio and IR can be confused with Infra Red. The title needs to the changed to A Novel Non-Contact Detection and Identification Method for Post-Disaster Compression State of Injured Individuals Using UWB Bio Radar and if the authors wish to distinguish further, then UWB [TD] where TD stands for Time Domain could be used.
The authors state "The electromagnetic waves in these frequency
bands have strong penetration capabilities through nonmetallic
materials" This statement needs to be qualified because at 7.29 GHz with a bandwidth of 1.4 GHz the loss in dB per metre of many materials such as concrete, bricks or natural material may be very high and losses in the region of 100's of dB/m can be expected, so the statement is not really credible.
There is no quantification of the loop gain of the radar, nor its minimum detectable signal level and this a a major weakness of the paper in that readers will have difficulty in understanding the fundamental performance of the radar. What is the effective radar cross section of the targets and when do the authors expect that targets will be become undetectable as a function of range, body mass and intervening materials? A graph might be helpful
The test results in section 4 do not provide information on the losses of the multiple layers, so the reader has no idea what is the real capability of the radar and its processing. The authors need to provide this infomation.
The compressive state is not well described. What does this mean in terms of the peak pressure applied to the subjects. Is this the distributed pressure over the body area or the peak pressure on say a small section of the rib cage of the male / female subjects and in relation to their individual body mass? The authors need to provide better information on this aspect.
Round 2
Reviewer 1 Report
Thank you for well revising the manuscript.
My concerns are very carefully addressed and i appreciate author efforts.
It would be great that authors revise the Figure 1 caption in a detail way as it is a main block for the paper.
N/A
Reviewer 3 Report
While the title has been changed there are still references within the paper to IR- UWB which need to be amended. For the record the main modulation schemes for UWB ranging radar are time domain, frequency domain and noise radar [or variants thereof]. Time domain UWB radar using sampling receivers has the lowest receiver sensistivity due to the wide receiver bandwidth, while frequency domain radars, stepped frequency or swept frequency have the best receiver sensisitivity due to the lower IF bandwidth and outperform sampling receivers by a factor of 20-40dB depending on design. Nosie radars can be sensitive but the correlation sidelobes reduce performance. The author's choice of timedomain and it is to be assumed sampling techniques do not provide the best receiver sensistivity. This should be reported in the paper.
The use of gypsum plaster board is such a low attenuation material that the authors need to explain that it offers the optimum test situation. The grpah shown by them in their response is questionable and wet concrete some building materials have significant losses above 1GHz. This should be reported to place the research into context.
Additionally real life earthquake created rubble is randonly orientated and is highly scattering and can increase path losses by tens of dB. This practical issue has caused previous researchers to encounter limitations to detectability and needs to be reported.
Minor
